# A New Approach for Classifying the Permanent Deformation Properties of Granular Materials under Cyclic Loading

**DOI:** 10.3390/ma16062141

**Published:** 2023-03-07

**Authors:** Gang Liu, Mingzhi Zhao, Qiang Luo, Jianchuan Zhou

**Affiliations:** 1School of Architecture and Civil Engineering, Xihua University, Chengdu 610039, China; 2MOE Key Laboratory of High-Speed Railway Engineering, Southwest Jiaotong University, Chengdu 610031, China

**Keywords:** granular materials, permanent strain rate, negative-power function, deformation tendency factor, steady-state line (SSL)

## Abstract

A series of medium-sized cyclic triaxial tests were performed to investigate the permanent deformation properties of granular materials. The strain rate was then plotted against loading cycles to classify the permanent deformation properties of granular materials under different cyclic stress ratios (CSRs). It was found that (1) the permanent strain rate dε_p_/dN was linearly correlated with loading cycles N using a double-log coordinate on the condition of CSR < 60%; (2) the deformation tendency factor β, which was extracted from the linear relationship between dε_p_/dN and N, significantly varied with CSR and, thus, can be adopted to identify the deformation states; (3) β > 1 implying that permanent strain accumulation ceases in limited cycles and corresponds to the plastic shakedown range, while 0 < β ≤ 1 indicates the temporary steady state, corresponding to the plastic creep range; (4) sluggish decrease or remarkable increase in dε_p_/dN appeared as CSR ≥ 60%, leading to soil collapsed in limited loading cycles and resulting in an incremental collapse range. The new approach was validated by the crushed tuff aggregates and subgrade materials reported previously. It is expected that the new approach will have wider applicability than the traditional one and can provide technical guidance for the design and construction of substructures in roadway and railway engineering.

## 1. Introduction

Granular materials are widely adopted to fill base and subbase layers in road pavement [1,2], ballast and sub-ballast in traditional railway [3,4], and subgrade layers in high-speed railway [5,6]. As the structure or substructure of road pavement and rail track, the granular functional layer suffers dynamic traffic load and transfers downward, playing a critical role in reducing and dispersing dynamic loads to avoid excessive deformation for the whole transport system. On the other hand, with the rapid development of transportation demand, the volume of traffic has become heavier, and the traffic speed has become higher for both highway and railway transportation, which presents new demands on the limitations of permanent deformation for the granular functional layers [7,8]. In particular, the post-construction deformation of high-speed railway is required and needs to be specified at millimeter-scale. Therefore, it is of great significance to investigate the permanent deformation properties of granular materials to gain an understanding of the service performance for the granular functional layer.

To evaluate the permanent deformation properties of granular materials, two kinds of technical approaches have been put forward in the literature. One is to develop an analysis model to predict the relationship between permanent deformation and loading cycles [9,10,11,12,13,14]. This approach has the purpose of calculating the exact deformation value of the granular materials. The other emphasizes quantitative evaluation of the developing tendency to permanent deformation [15,16,17,18]. In this framework, different deformation stages can be categorized in terms of cyclic stress level, the degree of compaction of granular materials and other external conditions. Since the definite value of millimeter-scale deformation is difficult to predict accurately, the latter approach has been promoted more and is more widely recognized.

The permanent deformation properties of granular materials are greatly influenced by the load amplitude in cyclic loading conditions. The tendency to develop deformation properties is mostly classified in terms of cyclic amplitude. Several classification methods for the long-term deformation properties of granular materials have previously been recommended, among which shakedown theory has received the most attention [19,20,21]. The shakedown concept was originally proposed to illustrate the properties of pressure vessels subjected to cyclic thermal loading. Later, it was used to analyze the characteristics of metal surfaces under repeated rolling load. This concept classified the behaviors of materials into four categories, including elastic shakedown, plastic shakedown, plastic creep and incremental collapse. In 1983, this concept was first introduced by Sharp to analyze the permanent deformation behavior of pavement materials [15]. Due to the existence of a post-compaction period, purely elastic deformation is not applicable for granular materials; therefore, the permanent deformation properties of the granular materials have been further grouped into plastic shakedown (Range A), plastic creep (Range B) and incremental collapse (Range C) by Werkmeister [17,22,23] based on empirical criteria:$$\mathrm{Range}\;A:\;\Delta \varepsilon_{p,5000}-\Delta \varepsilon_{p,3000}<4.5\times 10^{-5};\;\;\;\;\;\;\;\;\;\;\;\;\;\;\;\;\;\;\;\;\;\;\;\;\mathrm{Range}\;B:\;4.5\times 10^{-5}<\Delta \varepsilon_{p,5000}-\Delta \varepsilon_{p,3000}<4.0\times 10^{-4};\;\mathrm{Range}\;C:\;\Delta \varepsilon_{p,5000}-\Delta \varepsilon_{p,3000}>4.0\times 10^{-4}.$$
where Δ*ε_p_*_,3000_ and Δ*ε_p_*_,5000_ are the accumulated permanent strains at the 3000th and 5000th loading cycle, respectively. These classification criteria were also incorporated in European Standard EN-13286-7 [24]. In this framework, two critical stress levels, later termed the “plastic shakedown limit” and the “plastic creep limit”, were determined by the axial strain accumulated from the 3000th to 5000th loading cycles to distinguish the above-mentioned three ranges. Although the criteria are consistent with shakedown theory, it may not be applicable to all the granular materials in terms of grain size gradation and test conditions [19,20,25,26,27].

Gu et al. graded a series of flexible base materials in Texas according to Werkmeister’s criteria and the TxDOT specification (i.e., Item-247) [19,28]. Most of the tested materials were in Range C (i.e., incremental collapse) based on Werkmeister’s criteria, which indicates that these materials should be avoided as a pavement base course. Nevertheless, the test materials were mostly in Grade 2 in accordance with Item-247. This implies that the tested materials are recommended to fill a pavement base course. Since Item-247, as a local specification in Texas, is more instructive for grading the base course fillings, the Werkmeister’s criteria need to be adjusted to be consistent with Item-247. In order to be applicable to the Texas pavement materials, the shakedown boundaries were modified and are presented in Table 1. Wang and Zhuang performed a series of dynamic triaxial tests on subgrade materials under different loading frequencies, confining pressures and cyclic stress ratios [20]. They pointed out that the Werkmeister’s boundaries were not suitable for the tested crushed limestone aggregate. With this in mind, new boundaries were proposed for subgrade limestone materials, which are listed in Table 1.

As shown in Table 1, the shakedown and creep limits determined by Δ*ε_p_*_,5000_ − Δ*ε_p_*_,3000_ vary significantly with material type, size gradation and test conditions. It is quite difficult to classify the long-term deformation properties of granular materials under different cyclic stress levels by only calculating Δ*ε_p_*_,5000_ − Δ*ε_p_*_,3000_ since the choice of these two relevant strains (strains at the 3000th and 5000th loading cycles) are arbitrary to some extent [20,21]. In this regard, new criteria for grading the long-term deformation properties of granular materials are expected to have more general applicability. In this study, medium-sized triaxial tests were performed on a kind of graded gravel with different degrees of compaction (DoC) under different confining pressures. The relationship between the axial strain rate and the loading cycles of the gravel samples was presented in a double-log coordinate. The initial strain rate factor *α* and deformation tendency factor *β* were put forward to normalize the strain rate behaviors under different CSRs. A new approach for classifying the deformation properties of granular materials is proposed in terms of the correlation of the strain rate with loading cycles and the deformation tendency factor. The new criteria were further validated by the permanent deformation properties of other granular materials reported previously. It is expected that the new method will have wider applicability than the traditional method in terms of material types and test conditions.

## 2. Materials and Methods

### 2.1. Test Materials

The granular materials used in this study were a kind of graded gravel. They originated from quartz stone and sandstone. The materials were mechanically crushed from boulders and pebbles to form coarse particles with different shapes and sizes. After removal to the laboratory, the gravel samples were hand-washed with tap water to remove fine particles. After washing, the samples were placed in an oven and dried for 4 h at 105 °C to obtain the pure gravel samples. Then, the pure samples were sieved by square hole sieves for classification into eight groups, as shown in Figure 1. The grain size distribution of the graded gravels was redesigned to obtain favorable grading, as demonstrated in Figure 2. The upper and lower limits of the granular materials that were adopted to construct the subgrade in high-speed railway are also presented in Figure 2 as a reference in accordance with TB 10621-2014.

### 2.2. Test Apparatus

The triaxial test apparatus used in this study was designed and manufactured by GDS Instruments, Ltd., as shown in Figure 3. The confining pressure controller, which can provide the largest confining pressure of 2 MPa, is connected with a triaxial cell with a plastic tube to force de-aired water into the cell to impose a confining pressure. The actuating device at the bottom is designed in the loading process to apply axial cyclic stress to the gravel sample. At the same time, load and displacement transducers can collect data in real time automatically to feedback to the computer. The apparatus can perform cyclic triaxial tests for samples with a diameter of 150 mm. It is recommended by AASHTO T307-99 and GB T 50123-2019 that the ratio of the triaxial sample diameter to the maximum particle size should be not less than five to one [29,30]. Thus, the apparatus can only accommodate a soil sample with a maximum particle size of 30 mm. Since the maximum particle size of the tested gravel specimen was 31.5 mm, the particles of size larger than 30 mm were removed to fulfill the specification requirements. Although the scaling down method may alter the mechanical behaviors of the graded gravel slightly, the experimental results are still valuable for studying the long-term deformation properties of granular materials.

It is worth noting that an infinite volume controller (IVC) was adopted to supply de-aired water continuously with two confining pressure controllers connected on one side in this experiment, as it is difficult for a single confining pressure controller to provide sufficient de-aired water to the load cell to hold a constant confining pressure, as shown in Figure 3. At the same time, there were two water outlets/inlets on the other side. One of them was linked with the load cell and the other was connected with the water source. In the consolidated process, one pressure controller was activated to supply pressure water to the load cell to impose a confining pressure, and the other controller pumped water from the water source to fill the water tank. Once the first controller was exhausted, the IVC switched automatically to activate the second controller to supply water pressure. At the same time, the first controller was unblocked to the water source to replenish water into the controller. When the second controller was exhausted, the first controller was activated again. The two confining pressure controllers were activated alternatively to supply water until the pressure of water in the load cell was sufficient to provide a constant confining pressure. In this way, the confining pressure in the load cell was able to be kept constant.

### 2.3. Sample Preparation

A modified Proctor compaction test was performed on the graded gravel samples with water content of 0.2%, 2.1%, 3.8% and 5.4%. The test resulted in a maximum dry density of 2.43 g/cm^3^ and an optimum water content of 4.08%. The triaxial samples were intended to be 150 mm in diameter and 300 mm in height. The gravel samples were first oven-dried for 24 h and weighed to prepare the triaxial samples with degrees of compaction (DoC) of 0.95 and 1.00. After that, the granular samples were mixed with a suitable amount of de-aired water to reach the optimum water content (OWC) state and stored in a humidor for 24 h to homogenize the moisture. The OWC samples were then divided into five equal proportions to prepare the sample in five layers. Each layer remained the same in both height and weight.

The granular samples were compacted in a compaction container that was composed of a three-way split former. During compaction, each equally divided part of the sample was placed into the container and compacted by a Proctor hammer. For each tamping cycle, the hammer was set at the same initial height to ensure the same compaction energy each time. After being tamped to the desired height, the surface of the layer was scraped to a depth of 2 mm to ensure good interlocking with the adjacent layers [7,31]. Since the lower layers receive more compaction energy compared with the upper layers, this sample preparation method may inherently cause nonuniformity along the depth. However, this error can be neglected due to the same effect for each sample. After compaction, a rubber membrane was spread to enclose the sample and silicone grease was smeared across the top and bottom of the specimen to minimize friction with the top cap and bottom pedestal.

### 2.4. Test Program

Both monotonic and cyclic triaxial tests were conducted on gravel samples in the OWC state. The monotonic consolidated-drained triaxial test was performed first to obtain the peak shear strength of the gravel samples that can provide guidance for the loading scheme of the cyclic triaxial tests. The shearing velocity was set as low as 0.01 mm/min to minimize the pore pressure in the loading stage. The shearing process was ceased once the axial strain of the gravel samples reached 10%. Since granular materials are usually adopted to construct functional layers that are buried relatively shallowly, the peak shear strength (*σ*_1_ − *σ*_3_)*_f_* of the granular materials was measured on the condition that the confining pressure was set as 40 kPa and 60 kPa, respectively.

The cyclic tests were also performed in the drained condition. The sine-shaped waveform was selected to simulate the traffic loading in the cyclic triaxial test, as shown in Figure 4. The cyclic stress is denoted as *q^ampl^* and determined by the equation demonstrated as follows:(1)$$q^{ampl}=CSR\cdot {\left(\sigma_{1}-\sigma_{3}\right)}_{f}$$
where *CSR* is defined as the cyclic stress ratio, which is the ratio between the cyclic stress *q^ampl^* and the peak shear strength (*σ*_1_ − *σ*_3_)*_f_*. *CSR* varied from 5% to 80% in the cyclic triaxial tests.

In the cyclic test, the sample was first consolidated under a given confining pressure (i.e., 40 kPa or 60 kPa). Then, a small magnitude of deviatoric stress, which equaled 0.1 *q^ampl^*, was applied to the gravel sample to clear off the contact irregularities at the top of the specimen and to guarantee touch between the sample and the load head in the subsequent loading process. After that, a slow loading at a frequency of 0.1 Hz was first applied to the sample for 100 cycles to steady the sample. Then, the granular samples were loaded for another 9900 cycles at a frequency of 1 Hz. The application of the cyclic load lasted for approximately 3 h. The test data were recorded at 50 points for every loading cycle.

## 3. Test Results

### 3.1. Peak Shear Strength

The deviatoric stress was plotted against the axial strain for the granular samples with degrees of compaction (DoC) of 0.95 and 1.00, as shown in Figure 5. Figure 5a presents the stress–strain curves for the granular samples with DoC = 0.95. It can be seen that the deviatoric stress *σ*_1_ − *σ*_3_ of the gravel sample increased with axial strain *ε_a_* initially, and *σ*_1_ − *σ*_3_ reached a peak value of 605.5 kPa at *ε_a_* = 1.6% under the confining pressure *σ*_3_ of 40 kPa. Then, *σ*_1_ − *σ*_3_ began to decrease and finally leveled off to its residual level once *ε_a_* went beyond 1.6%. The stress-strain curve under *σ*_3_ = 60 kPa presented the same features as that under *σ*_3_ = 40 kPa, and the peak shear strength (*σ*_1_ − *σ*_3_)*_f_* reached 713.4 kPa at *ε_a_* = 1.6%. In other words, the tested granular samples with DoC = 0.95 demonstrated strain-softening behaviors under the conditions of *σ*_3_ = 40 kPa and *σ*_3_ = 60 kPa.

Similarly, the shear behavior of the granular samples with DoC = 1.00 also demonstrated strain-softening features, as presented in Figure 5b. The peak shear strengths (*σ*_1_ − *σ*_3_)*_f_* reached 855.9 kPa and 955.5 kPa, respectively, for the samples under confining pressures of 40 kPa and 60 kPa. Moreover, it was interesting to find that the peak shear strengths for the samples with DoC = 1.00 were also obtained at *ε_a_* around 1.6%.

### 3.2. Permanent Deformation

During the cyclic loading process, both the pore pressure and axial deformation were monitored in real time to record the long-term properties of the granular samples. Due to the drained condition and low loading frequency, the pore water pressure stayed below 2 kPa, indicating that the influence of the pore water pressure could be neglected. The granular samples presented quasi-sine-shaped deformation curves due to the sine-shaped cyclic stress. Both elastic and permanent deformation occurred in each loading cycle. The elastic deformation was recovered repeatedly over a large number of loading cycles, while the permanent deformation was accumulated gradually. The permanent strain was plotted against the loading cycles to illustrate the long-term deformation properties of the tested granular samples. Figure 6 shows the relationship between the permanent strain *ε_p_* and the loading cycles *N* of the gravel samples with DoC = 0.95.

In Figure 6a, with a confining pressure of 40 kPa, the permanent axial strain *ε_p_* was accumulated to less than 1% in 10,000 loading cycles for the samples with CSR < 60%. The permanent strain *ε_p_* mostly developed at the initial stage of the loading process. As CSR increased to 60%, *ε_p_* reached 2% at the 10,000th loading cycle, and significant accumulation of permanent strain still occurred at the end of the loading process. This implied that additional strain was expected to develop continuously with loading cycles until the granular sample collapsed. For CSR = 80%, the axial strain was accumulated in an accelerated manner in the first few hundred loading cycles and the granular sample collapsed momentarily. Figure 6b depicts the permanent strain of the granular samples with DoC = 0.95 under the confining pressure of 60 kPa. It was observed that the samples presented quite similar deformation properties at the confining pressure of 60 kPa to those at the confining pressure of 40 kPa. It is remarkable that an increase in cyclic stress and CSR was able to speed up the development of the permanent strain of the granular materials and exacerbate the service performance of the transport infrastructure.

The relationship between permanent strain *ε_p_* and loading cycles *N* for samples with DoC = 1.00 is demonstrated in Figure 7. Similarly, *ε_p_* of the granular samples was accumulated to a small magnitude on the condition of CSR < 60%. However, the strain developed continuously, or even in an accelerated manner, once CSR reached 60% and 80%.Evidently, the permanent axial strain exhibited similar response behaviors under the effect of cyclic stress, regardless of the confining pressure and the degree of compaction (DoC).

### 3.3. Permanent Deformation Rate

The strain rate d*ε_p_*/d*N* of each tested granular sample can be calculated from the relationship between the permanent strain *ε_p_* and the loading cycles *N*. d*ε_p_*/d*N* was plotted against *N* using a double-log coordinate, as shown in Figure 8. Figure 8a demonstrates the permanent strain rate development characteristics for samples with DoC = 0.95 under the confining pressure of 40 kPa. It can be seen that d*ε_p_*/d*N* had a linear relationship with *N* on a log-log scale on the condition of CSR < 60%. Moreover, it was observed that d*ε_p_*/d*N* attenuated more quickly at lower CSRs than that at higher CSRs, resulting in a steeper slope of strain rate curves at lower CSRs. For CSR = 60%, despite the fact that d*ε_p_*/d*N* decreased almost linearly with *N* using a double-log coordinate in the first 1000 loading cycles, the attenuation of the strain rate occurred slowly in the subsequent loading cycles. It can be deduced that d*ε_p_*/d*N* would finally remain steady at a constant level or may even increase with loading cycles *N* at this cyclic load amplitude. Once CSR reached 80%, d*ε_p_*/d*N* started to increase with *N* after approximately the 500th loading cycle, although it decreased initially. This implies that the sample would collapse rapidly in limited loading cycles.

Figure 8b–d show the relationships between the permanent strain rate d*ε_p_*/d*N* and loading cycles *N* at different CSRs for samples with other DoC or confining pressures. It is interesting to note that the rate curves presented almost the same features as those demonstrated in Figure 8a, indicating that the strain rate development features were only susceptible to CSR, and were insensitive to the degree of compaction (DoC) and the confining pressure.

## 4. Discussion

### 4.1. Modelling and Normalizing the Strain Rate Development

The permanent strain rate d*ε_p_*/d*N* was used in this study to classify the permanent deformation properties of granular soils, since the strain rate is the key factor for determining the deformation tendencies. As mentioned above, the permanent strain rate d*ε_p_*/d*N* was linearly correlated with the loading cycles using a double-log scale on the condition of CSR < 60%; therefore, d*ε_p_*/d*N* can be expressed by the following equation:(2)$$\log\frac{d\varepsilon_{p}}{dN}=-\beta \log N+\alpha$$
where *α* is related to the initial strain rate, and equals the logarithm of the strain rate at the first loading cycle in value, and *β* represents the development tendencies of the permanent deformation for granular materials. Both *α* and *β* are constants that vary with CSRs for a given sample under a given confining pressure. Therefore, the parameter *α* under different CSRs can be calculated by the intercept of log(d*ε_p_*/d*N*)-log*N* curves, and the parameter *β* can be evaluated by the slope of the segment from the 100th to the 10,000th loading cycle of log(d*ε_p_*/d*N*)-log*N* curves, as shown by the following equations:(3)$$\alpha ={\left(\log\frac{d\varepsilon_{p}}{dN}\right)}_{N=1}$$
(4)$$\beta =\frac{1}{2}\left[{\left(\log\frac{d\varepsilon_{p}}{dN}\right)}_{N=100}\;-\;{\left(\log\frac{d\varepsilon_{p}}{dN}\right)}_{N=10,000}\right]$$
where ${\left(\log\frac{d\varepsilon_{p}}{dN}\right)}_{N=1}$, ${\left(\log\frac{d\varepsilon_{p}}{dN}\right)}_{N=100}$ and ${\left(\log\frac{d\varepsilon_{p}}{dN}\right)}_{N=10,000}$ are the permanent strain rates of the tested samples at the 1st, 100th and 10,000th loading cycles. In this regard, the initial strain rate factor *α* and the deformation tendency factor *β* under different CSRs were calculated for the samples with different DoC and confining pressures, as shown in Figure 9 and Figure 10.

As shown in Figure 9, the initial strain rate factor *α* increased with CSR for samples with a given DoC under a given confining pressure, indicating that the strain rate in the initial stage rose remarkably with cyclic load amplitude. Figure 10 demonstrates that the deformation tendency factor *β* decreased with increase in cyclic load amplitude. Moreover, there was an approximately linear relationship between *β* and CSR. This implies that the strain rate attenuated more rapidly against loading cycles at lower CSRs. With increase in CSR, the decrease in the strain rate against loading cycles gradually became sluggish. As the CSR reached 60% or 80%, d*ε_p_*/d*N* of the tested samples decreased with loading cycles initially, but the decreasing tendency gradually weakened with loading cycles. For CSR = 80% granular samples, d*ε_p_*/d*N* increased after approximately the 500th loading cycle. In other words, there was an inflection point in the strain rate-loading cycles curve; therefore, the whole segment of the curve was not suitable to be represented by Equation (2).

Since the initial strain rate factor *α* and the deformation tendency factor *β* are critical parameters to depict the strain rate development under the condition of CSR < 60%, a new indicator to characterize the strain rate of granular materials with loading cycles can be put forward and shown as follows:(5)$$I_{d\varepsilon /dN}=-\frac{\log\left(d\varepsilon_{p}/dN\right)-\alpha }{\beta }$$
where *I*_d*ε*/d*N*_ is the new indicator that can be termed the strain rate index. In accordance with this equation, the relationship between d*ε_p_*/d*N* and *N* can be transformed into the correlation of *I*_d*ε*/d*N*_ with *N*. It was interesting to find that *I*_d*ε*/d*N*_ was linearly correlated with *N* using a semi-log scale under CSR < 60% for the tested granular samples. More importantly, the relationship between *I*_d*ε*/d*N*_ and *N* of the samples with different CSRs, DoC and confining pressures developed almost along the same line. A linear regression analysis was performed with the following equation producing a correlation coefficient as high as 0.99:(6)$$I_{d\varepsilon /dN}=0.85\log N$$

The fitting line, together with the tested *I*_d*ε*/d*N*_ of the samples, are presented in Figure A1 in Appendix A. Figure 11 shows a comparison of the fitting line and the tested *I*_d*ε*/d*N*_ for the samples with different DoC and confining pressures. *I*_d*ε*/d*N*_ was linearly correlated with *N* using a semi-log scale under different CSRs for samples with DoC = 0.95 under the confining pressure of 40 kPa [Figure 11a]. This means that the correlation of d*ε_p_*/d*N* with *N* under different CSRs can be normalized by transforming into *I*_d*ε*/d*N*_. Interestingly, the *I*_d*ε*/d*N*_-log*N* curves for samples with other DoC and confining pressures can also be normalized on the condition of CSR < 60%, as demonstrated in Figure 11b–d; the normalized equation remains the same and can also be presented by Equation (6). Therefore, the line determined by Equation (6) can be termed the steady state line (SSL). In other words, the strain rate of the sample with CSR < 60% showed a steady attenuation, which indicates that the sample maintained a steady state, permanently or temporarily.

It is worth noting that the strain rate curves for the samples under CSR = 60% or 80% could not be normalized due to the difficulty in determining *α* and *β*. More importantly, given the fact that the deceasing trend of the strain rate weakened remarkably and an increasing trend emerged in the limited loading cycles for samples under CSR of 60% and 80%, the granular samples would undoubtedly be damaged in subsequent loading cycles. The sluggish attenuation or increase in the potential of the strain rate implies that the sample collapsed in limited cycles, and that the deformation features can be defined as incremental collapse. As for the samples under CSR < 60%, the strain rate was linearly decreased with loading cycles using a double-log scale, indicating that the samples stayed in a steady state under the effect of limited loading cycles. Furthermore, two critical factors *α* and *β* can be put forward to normalize the strain rate behaviors for samples under CSR < 60%. This implies that the sample with CSR < 60% can remain steady in limited loading cycles.

### 4.2. Critical State Analysis

Although all the strain rate curves can be normalized by the strain rate index *I*_d*ε*/d*N*_ for samples under CSR < 60%, this does not imply that all the samples will not collapse all along. The expression to model the relationship between the permanent strain rate d*ε_p_*/d*N* and the loading cycles *N*, i.e., Equation (2), can be rewritten as follows:(7)$$\frac{d\varepsilon_{p}}{dN}=10^{\alpha }N^{-\beta }$$

Moreover, the permanent strain can be evaluated from the integral form of Equation (7), as shown by the following equation:(8)$$\left\{\begin{array}{l} \varepsilon_{p}=\frac{10^{\alpha }}{1-\beta }N^{1-\beta }+C,\;\;\;\;\;\;\left(\beta \neq 1\right)\\ \varepsilon_{p}=10^{\alpha }\ln N+C,\;\;\;\;\;\;\left(\beta =1\right) \end{array}\right.\;\;\;\;\;$$
where *C* is a constant that varies with the samples. Equation (8) is applicable to predict the permanent strain of the granular soils under cyclic loading with *C* determined by the measured strains. In this regard, *β* = 1 is a critical state to determine the deformation tendencies of granular materials. The situation *β* > 1 denotes that the permanent strain *ε_p_* will finally converge to a constant value with loading cycle *N* becoming infinitely great, and the soil sample can remain steady under a cyclic loading effect. As 0 < *β* ≤ 1, *ε_p_* increases continuously with *N*. This implies that the soil will collapse sooner or later with *N* continuously increasing.

The sample with DoC = 0.95 under the confining pressure of 40 kPa can be selected to predict the permanent strain development at millions of loading cycles in terms of Equation (8), as shown in Figure 12. It can be seen that the permanent strain levels off to a relatively small magnitude of constant value after approximately the 10 millionth loading cycle on the condition of *β* < 1. In the subsequent loading cycles, the development of permanent strain is ceased, indicating that only elastic deformation occurs for the granular soil. Furthermore, the situation of *β* = 1.137, which denotes that the deformation tendency factor *β* is only slightly greater than unity, finally results in a small magnitude of permanent strain of 0.11% at the 200 millionth loading cycle. On the other hand, the permanent strain develops continuously with loading cycles once *β* goes below unity. For example, the soil sample has permanent strains of 1.07%, 1.48% and 1.63% at the 10 millionth, 100 millionth and 200 millionth loading cycles as *β* decreases to 0.876 at a targeted CSR of 40%. This implies that a significant increase in permanent strain still occurs after the 200 millionth loading cycle. On the other hand, the strain has already increased to 1.63%, which is a relatively high magnitude, on the condition of *β* = 0.876. Note that the strain corresponding to the peak shear strength in the monotonic shear test is approximately 1.6%, and the granular sample will undoubtedly collapse if the loading cycles continuously increase.

From the discussion above, the granular samples under CSR of 60% and 80% had a strain rate that showed a sluggish decrease or remarkable increase over limited loading cycles. It is not difficult to deduce that the granular sample will be damaged in a limited time; therefore, the samples with CSR = 60–80% can be regarded as falling into an “incremental collapse” state. As for the granular samples with CSR < 60%, the strain rate decreased regularly when plotted against the loading cycles using a double-log coordinate. The permanent strain rate development can be represented by a negative power function; the deformation tendency factor *β* is put forward as the critical parameter to classify the permanent deformation properties of granular soils under CSR < 60%. As a result, the samples with CSR ≤ 20% had *β* > 1, indicating that accumulation of permanent strain ceased in a few loading cycles, and the sample was able to remain in a permanent steady state (PSS) in spite of infinite loading cycles. This state was termed “plastic shakedown” by Werkmeister [22,23]. The samples with 20% < CSR < 60% led to *β* ≤ 1. This indicates that permanent strain increased continuously with loading cycle until the soil became collapsed, though the strain rate developed along SSL in limited loading cycles. Therefore, *β* ≤ 1 represents a temporarily steady state (TSS) of the sample and corresponds to the “plastic creep” range.

### 4.3. Verification of the Newly Proposed Criteria

The difference between the accumulated permanent strains at the 3000th and 5000th loading cycles, i.e., Δ*ε_p_*_,5000_ − Δ*ε_p_*_,3000_, have been widely adopted to classify the permanent deformation properties of the granular materials. Werkmeister (2003) first classified the permanent deformation properties of granular materials into three categories in accordance with Δ*ε_p_*_,5000_ − Δ*ε_p_*_,3000_ [22,23]. Subsequently, some researchers proposed modified values for the plastic shakedown limit and the plastic creep limit within the Δ*ε_p_*_,5000_ − Δ*ε_p_*_,3000_ framework. However, the selection of Δ*ε_p_*_,5000_ − Δ*ε_p_*_,3000_ may be somewhat arbitrary, resulting in critical limits that may vary with material type and test conditions. In this regard, a new approach to categorize the permanent deformation properties of granular materials under different CSRs is proposed in this study. In this newly proposed approach, whether the relationship between d*ε_p_*/d*N* and *N* using a double-log scale stays linear or concaves upwards is adopted to identify the incremental collapse range, and *β* = 1 is regarded as the critical state to categorize the plastic shakedown and plastic creep ranges. The new criteria are expected to have wider applicability in terms of material types and test conditions.

The permanent deformation properties of crushed tuff aggregates under different cyclic amplitudes were investigated at both optimum and saturated water contents [7]. Figure 13 presents the development features of permanent deformation for saturated samples. The ratio of cyclic stress *q^ampl^* to confining pressure *σ*_3_, which is denoted as *q^ampl^*/*σ*_3_, was adopted to depict the variation of cyclic amplitudes. It can be seen that the crushed tuff aggregates demonstrated different development features for both permanent strain and strain rate under different cyclic amplitudes, as demonstrated in Figure 13a,b. Especially, as *q^ampl^*/*σ*_3_ varied from 1 to 5, the permanent strain rate d*ε_p_*/d*N* decreased linearly with *N* on the double-log coordinate. However, d*ε_p_*/d*N* presented a remarkable increase from the beginning of the 1000th loading cycle on the condition of *q^ampl^*/*σ*_3_ = 7.5, which corresponded to the incremental collapse range.

The deformation tendency factor *β* was evaluated in accordance with the linear relationship between log(d*ε_p_*/d*N*) and log*N*, and further plotted against *q^ampl^*/*σ*_3_, as shown in Figure 13c. Evidently, *β* was negatively correlated with *q^ampl^*/*σ*_3_ for the saturated samples, which remained the same with the gravel materials investigated in this study. As *q^ampl^*/*σ*_3_ increased from 1 to 5, *β* decreased from 1.304 to 0.953. The plastic shakedown and plastic creep can be identified with *β* = 1 as the critical state. Therefore, the plastic shakedown limit lies between *q^ampl^*/*σ*_3_ = 3 and 5, while the plastic creep limit is located between *q^ampl^*/*σ*_3_ = 5 and 7.5. These classification results are consistent with those demonstrated by Cao et al. (2017) [7], verifying the applicability of the newly proposed criteria to the saturated crushed tuff aggregates. Similarly, the strain rate d*ε_p_*/d*N* can be transformed into a strain rate index *I*_d*ε*/d*N*_, which can be further plotted against *N* to obtain the normalized SSL. Interestingly, the normalized SSL of the saturated crushed tuff aggregate remained the same as the SSL of the gravel samples in this study, as presented in Figure 13d.

Furthermore, the deformation properties of crushed tuff aggregates at optimum water content (OWC) are demonstrated in Figure 14. The development characteristics of the permanent strain and strain rate are shown in Figure 14a,b. The permanent strains of the OWC samples accumulated more sluggishly than those of the saturated samples under a given cyclic amplitude. The strain rate d*ε_p_*/d*N* was also remarkably lower for the OWC samples compared with the saturated one. In detail, d*ε_p_*/d*N* was linearly correlated with loading cycles *N* using a double-log scale with *q^ampl^*/*σ*_3_ varying from 1 to 7.5. This implies that the OWC samples will not enter into an incremental collapse range despite *q^ampl^*/*σ*_3_ reaching 7.5, which is different from those under saturated condition.

The deformation tendency factor *β* was then plotted against *q^ampl^*/*σ*_3_ for the OWC tuff samples, as presented in Figure 14c. The critical state *β* = 1 was again used to identify the plastic shakedown and plastic creep range, resulting in the plastic shakedown limit being located between 5 and 7.5. Since the increase in *q^ampl^*/*σ*_3_ to 7.5 could not lead to incremental collapse of the OWC sample, the plastic creep limit should be greater than 7.5. The values of these two critical limits obtained from the new criteria are consistent with those determined by Δ*ε_p_*_,5000_ − Δ*ε_p_*_,3000_. Finally, the strain rate curves under different cyclic amplitudes can be normalized into SSL when expressed by *I*_d*ε*/d*N*_, as shown in Figure 14d. Therefore, the permanent deformation properties of crushed tuff aggregates, at optimum or saturated water content, could be categorized in terms of the new criteria proposed in this study.

Furthermore, the permanent strain properties of the subgrade materials were classified by Wang and Zhuang (2021) with Δ*ε_p_*_,5000_ − Δ*ε_p_*_,3000_ as the criterion [20]. However, they point out that the plastic shakedown limit and creep limit proposed by Werkmeister (2003) [22] are inapplicable to the investigated subgrade materials. Thereby it is believed that the plastic shakedown range of the subgrade materials corresponds to Δ*ε_p_*_,5000_ − Δ*ε_p_*_,3000_ < 1 × 10^−5^, and the plastic creep corresponds to 1 × 10^−5^ < Δ*ε_p_*_,5000_ − Δ*ε_p_*_,3000_ < 8 × 10^−5^, while the incremental collapse corresponds to Δ*ε_p_*_,5000_ − Δ*ε_p_*_,3000_ > 8 × 10^−5^. Then, the development features of the permanent axial strain of the subgrade materials were investigated under different cyclic stress ratios (CSRs). It was found that the subgrade materials, with an effective confining pressure of 20 kPa and a loading frequency of 1 Hz, fell into the plastic shakedown range on the condition of CSR = 10% and 30% and entered the plastic creep range as CSR increased to 40% and 60%. Therefore, the plastic shakedown limit lies between CSR = 30% and 40% with Δ*ε_p_*_,5000_ − Δ*ε_p_*_,3000_ as the classification criterion.

On the other hand, the permanent strain rate d*ε_p_*/d*N* of the subgrade materials can be calculated from the relationship between the permanent strain *ε_p_* and the loading cycles *N*. Figure 15a shows the relationship between d*ε_p_*/d*N* and *N* for samples with an effective confining pressure of 20 kPa and a loading frequency of 1 Hz. Furthermore, the correlation of the deformation tendency factor *β* with CSR can be established, as shown in Figure 15b. It is obvious that the strain rate of the subgrade materials is also linearly correlated with loading cycles using a double-log scale, indicating that the relationship can be depicted by Equations (2) and (7). Therefore, *β* can be obtained for the samples under different CSRs. It is interesting to note that *β* was also negatively correlated with CSR for the subgrade materials, which remained the same with that investigated in this study. With *β* = 1 as the critical state, the plastic shakedown range and the plastic creep range can be categorized. CSR = 10% and 30% resulted in *β* greater than unity, which indicates that the sample remained steady all along and stayed in the plastic shakedown range. As CSR reached 40% and 60%, the strain rate development led to *β* < 1, corresponding to the plastic creep range. It is remarkable that the new criteria lead to the same classification results with *β* as the indicator. Moreover, the plastic shakedown limit of the subgrade materials corresponds approximately to CSR = 30–40%, whereas the shakedown limit of the granular materials investigated in this study corresponds to CSR that varies from 20% to 30%. The difference in the critical CSR may be induced by the test conditions. The cyclic triaxial tests were performed in an undrained condition by Wang and Zhuang (2021) [20], but the granular samples adopted in this study stayed in a drained condition in the whole loading process.

Furthermore, the difference value of the accumulated axial strains at the 3000th and 5000th loading cycles for the investigated samples in this study, i.e., Δ*ε_p_*_,5000_ − Δ*ε_p_*_,3000_, was recorded and listed in Table 2. The classification of the permanent deformation properties by the newly proposed criteria is also listed in the table. With the shakedown ranges determined by the new criteria, both the plastic shakedown limit and the plastic creep limit, if expressed by Δ*ε_p_*_,5000_ − Δ*ε_p_*_,3000_, can be obtained. The plastic shakedown limit should be 2.1 × 10^−4^~2.4 × 10^−4^, and the plastic creep limit located between 5 × 10^−4^ and 19 × 10^−4^. Remarkably, the definite values of both limits for the tested samples in this study were quite different from those in previous reports, as shown in Table 1. This further implies that the shakedown boundaries determined by Δ*ε_p_*_,5000_ − Δ*ε_p_*_,3000_ vary with the tested materials and experimental conditions. However, the critical state defined by *β* = 1 is applicable to the granular samples in this study, the crushed tuff aggregates used by Cao et al. (2017) [7] and the subgrade materials adopted by Wang and Zhuang (2021) [20]. Of course, the application scope of the new criteria needs more experimental verification.

## 5. Conclusions

Medium-sized cyclic triaxial tests were performed on gravel samples with different DoC under different confining pressures to investigate the permanent deformation properties. Some significative conclusions can be drawn as follows:(1)The development features of the permanent strain rate against the loading cycles can be adopted to identify the incremental collapse range for the granular materials. The permanent strain rate linearly decreases with loading cycles using a double-log scale on the condition of CSR < 60%. Once CSR reaches or goes beyond 60%, the strain rate increases with cycle numbers, inducing soil collapse in limited loading cycles and corresponds to the incremental collapse range.(2)A negative-power function can be adopted to characterize the relationship between the permanent strain rate of the granular samples and the loading cycles on the condition of CSR < 60%. The exponent *β* in the equation is a tendency indicator that can be selected to identify the long-term deformation properties of granular materials.(3)*β* = 1.0 is the critical state to determine the boundary of the plastic shakedown range and the plastic creep range. As *β* > 1.0, permanent strain accumulation can be ceased in limited loading cycles, which corresponds to the plastic shakedown range. Conversely, the permanent strain will increase continuously with loading cycles on the condition of *β* ≤ 1.0 and lead to soil collapse, corresponding to the plastic creep range.(4)The classification criteria proposed in this study were established based on the correlation of the permanent strain rate with the loading cycles using a double-log coordinate and the deformation tendency factor *β*. The newly proposed criteria are unaffected by material type and test conditions and are expected to have wider applicability than those in accordance with Δ*ε_p_*_,5000_ − Δ*ε_p_*_,3000_.

## Figures and Tables

**Figure 1 materials-16-02141-f001:**
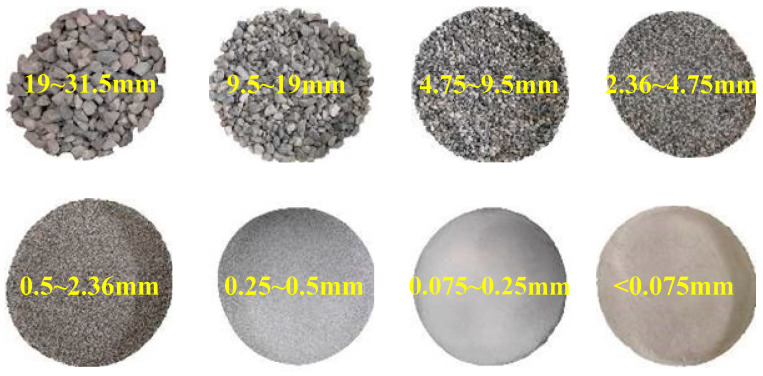
Granular particles with different grain sizes.

**Figure 2 materials-16-02141-f002:**
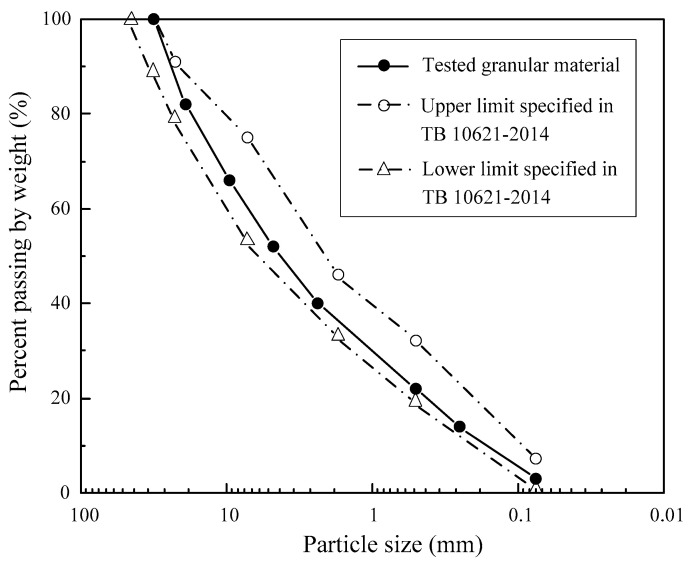
Grain size distribution of the graded gravel in this study.

**Figure 3 materials-16-02141-f003:**
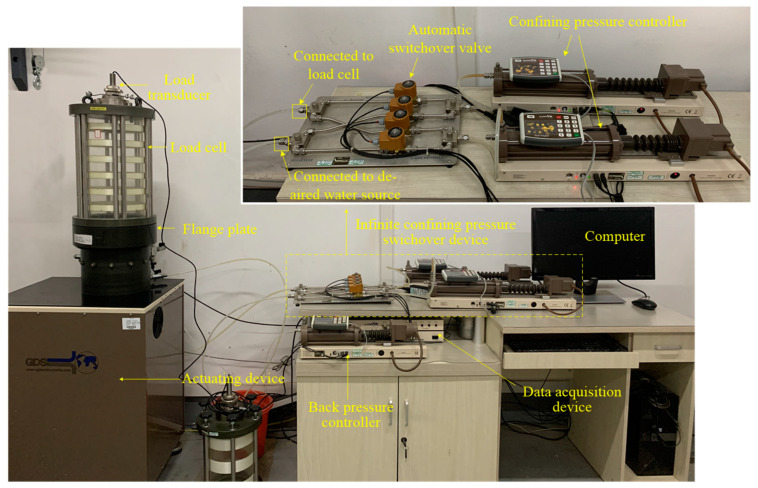
GDS triaxial test apparatus in this study.

**Figure 4 materials-16-02141-f004:**
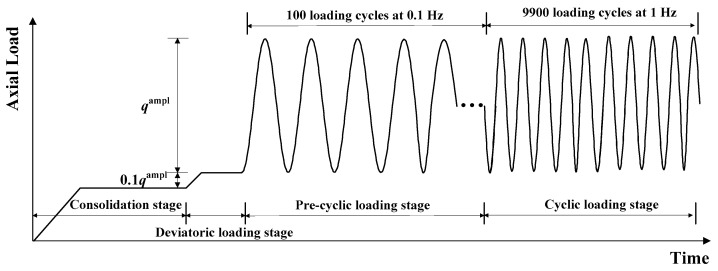
Cyclic loading program.

**Figure 5 materials-16-02141-f005:**
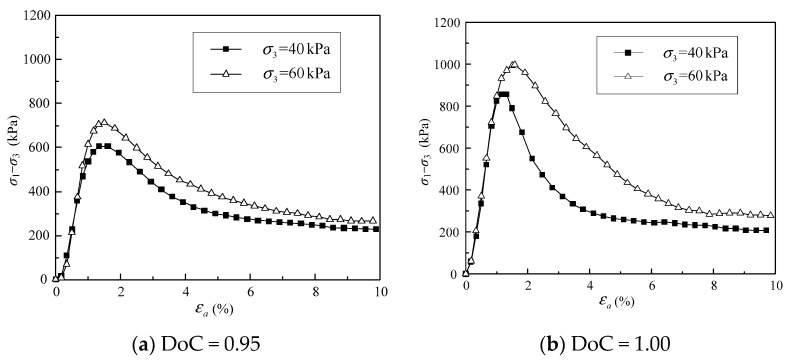
The relationship between deviatoric stress and axial strain for samples with (**a**) DoC = 0.95; and (**b**) DoC = 1.00.

**Figure 6 materials-16-02141-f006:**
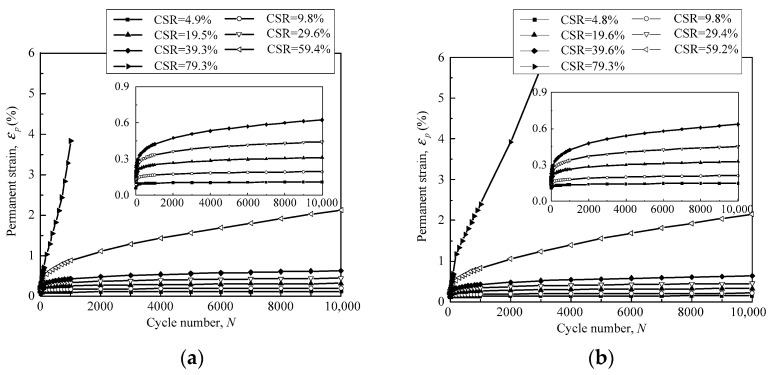
The relationship between permanent strain and loading cycles of granular samples with DoC = 0.95 under the confining pressure of (**a**) 40 kPa; and (**b**) 60 kPa.

**Figure 7 materials-16-02141-f007:**
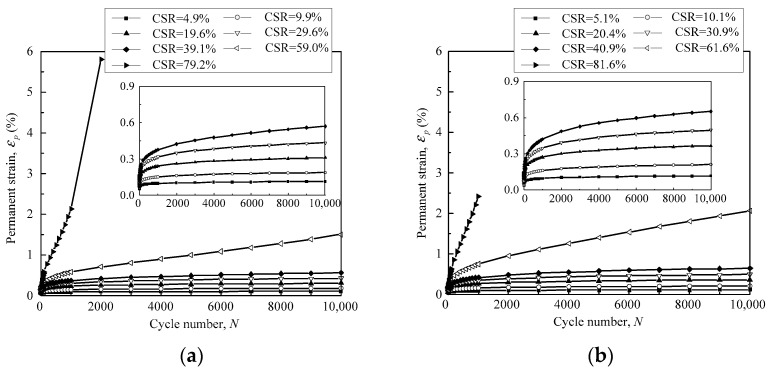
The relationship between permanent strain and loading cycles of granular samples with DoC = 1.00 under the confining pressure of (**a**) 40 kPa; and (**b**) 60 kPa.

**Figure 8 materials-16-02141-f008:**
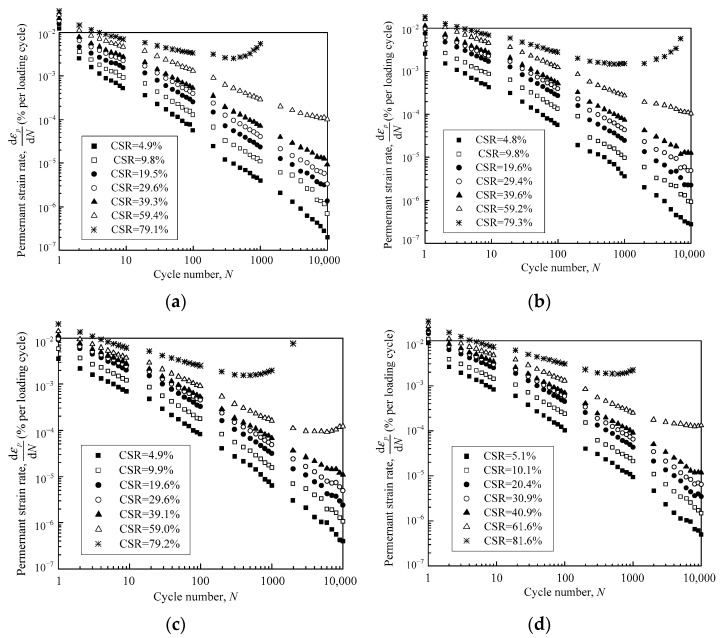
The relationships between strain rate and loading cycles under different CSRs for the tested samples with (**a**) DoC = 0.95, *σ*_3_ = 40 kPa; (**b**) DoC = 0.95, *σ*_3_ = 60 kPa; (**c**) DoC = 1.00, *σ*_3_ = 40 kPa; and (**d**) DoC = 1.00, *σ*_3_ = 60 kPa.

**Figure 9 materials-16-02141-f009:**
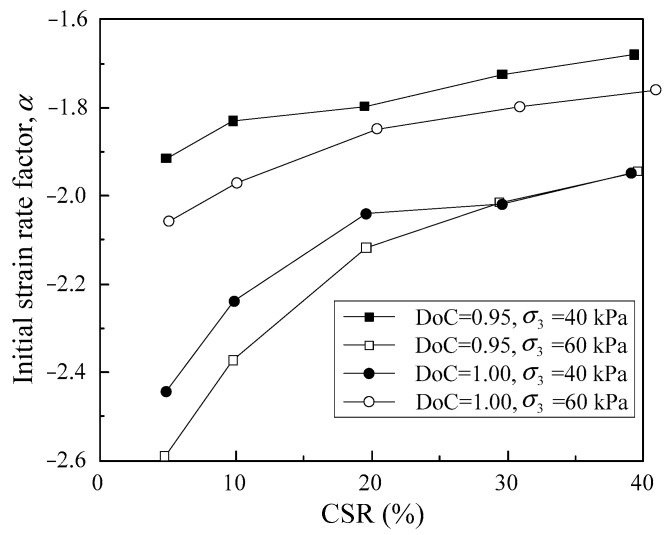
The correlation of initial strain rate factor *α* with CSR.

**Figure 10 materials-16-02141-f010:**
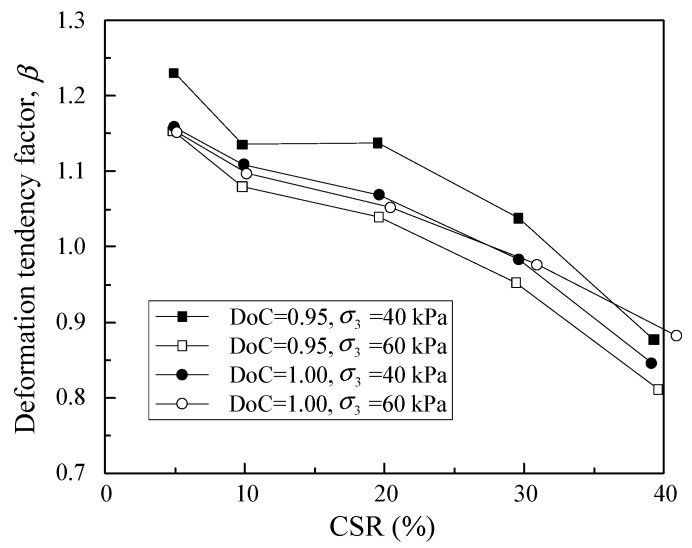
The correlation of deformation tendency factor *β* with CSR.

**Figure 11 materials-16-02141-f011:**
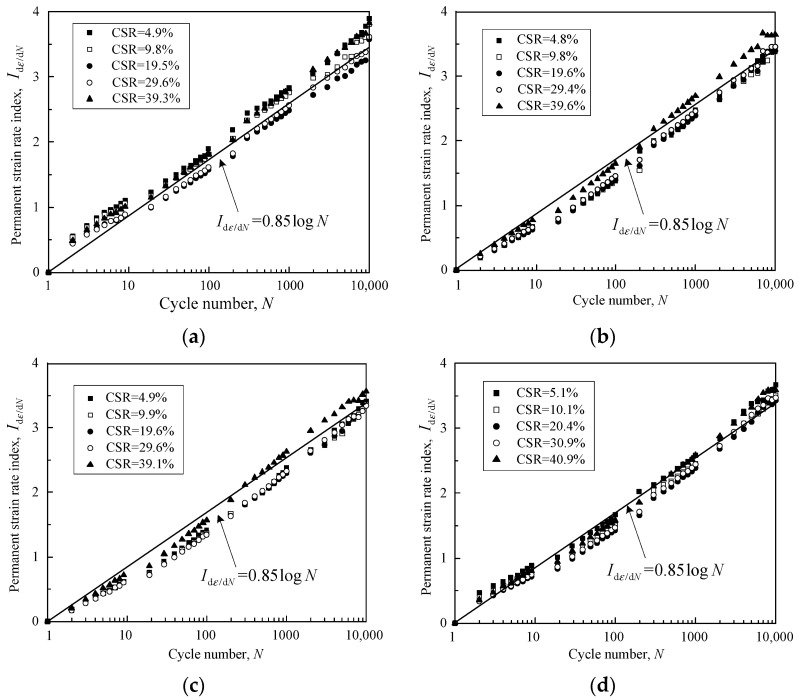
The relationship between strain rate index and loading cycles for the granular samples with (**a**) DoC =0.95, *σ*_3_ = 40 kPa; (**b**) DoC = 0.95, *σ*_3_ = 60 kPa; (**c**) DoC = 1.00, *σ*_3_ = 40 kPa; and (**d**) DoC = 1.00, *σ*_3_ = 60 kPa.

**Figure 12 materials-16-02141-f012:**
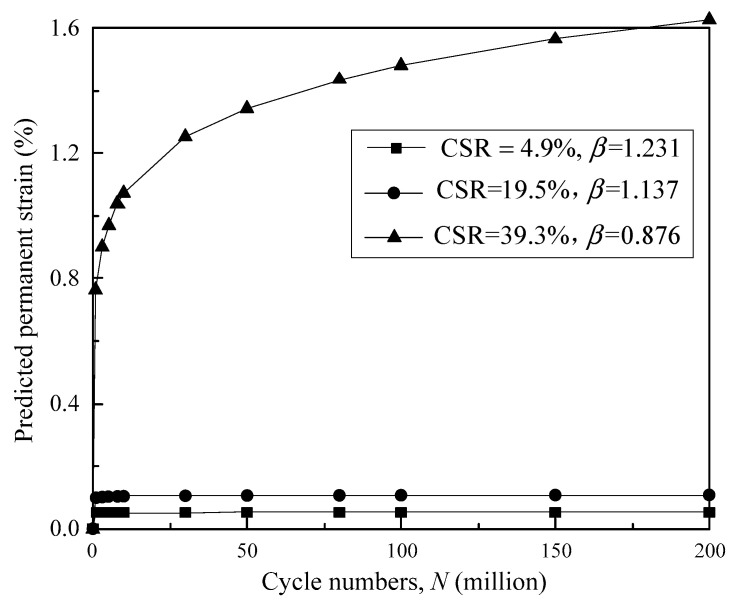
Predicted permanent strain at millions of loading cycles.

**Figure 13 materials-16-02141-f013:**
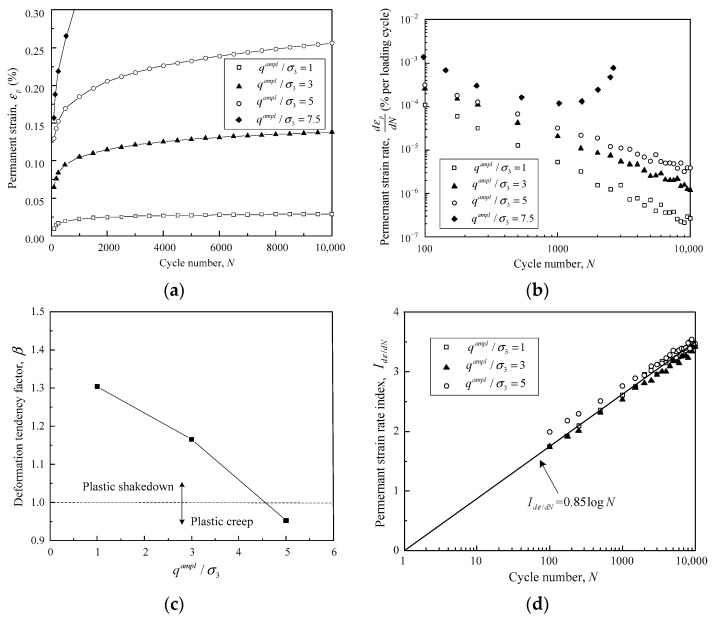
Permanent deformation properties of crushed tuff aggregates at saturated water content: (**a**) the relationship between permanent strain and loading cycles; (**b**) permanent strain rate plotted against loading cycles in double-log coordinate; (**c**) development of deformation tendency factor *β* against cyclic amplitude; (**d**) normalized strain rate curves under different cyclic amplitudes. (Data calculated from Cao et al., 2017) [7].

**Figure 14 materials-16-02141-f014:**
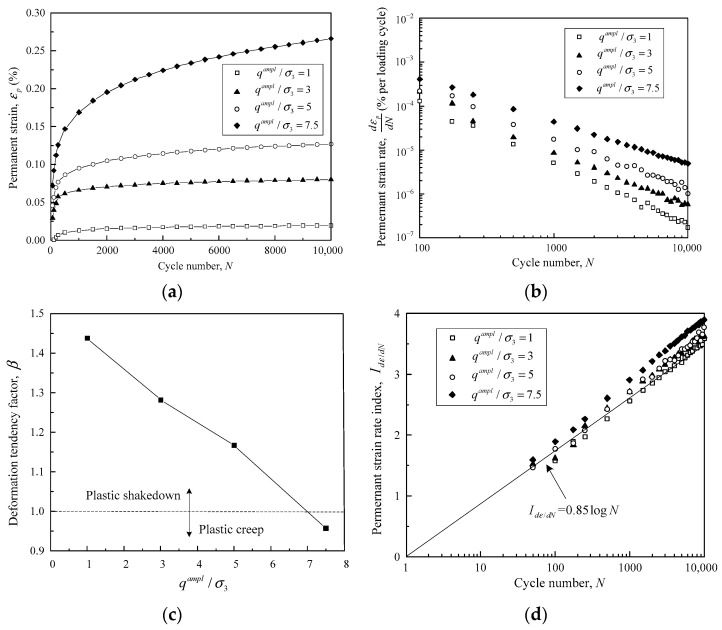
Permanent deformation properties of crushed tuff aggregates at optimum water content: (**a**) the relationship between permanent strain and loading cycles; (**b**) permanent strain rate plotted against loading cycles in double-log coordinate; (**c**) development of deformation tendency factor *β* against cyclic amplitude; (**d**) normalized strain rate curves under different cyclic amplitudes. (Data calculated from Cao et al., 2017) [7].

**Figure 15 materials-16-02141-f015:**
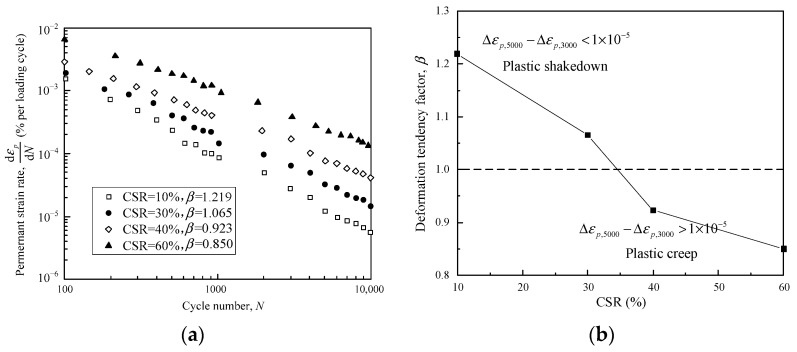
Permanent deformation properties of subgrade materials: (**a**) permanent strain rate plotted against loading cycles in double-log coordinate; (**b**) the relationship between *β* and CSR. (Data calculated from Wang and Zhuang, 2021) [20].

**Table 1 materials-16-02141-t001:** The shakedown and creep limits determined by Δ*ε_p_*_,5000_ − Δ*ε_p_*_,3000_ from different studies.

No.	Plastic Shakedown Limit	Plastic Creep Limit	Filling Position	Literature
1	4.5 × 10^−5^	4.0 × 10^−4^	Pavement	Werkmeister 2003, 2006 [22,23]
2	6.0 × 10^−5^	6.0 × 10^−4^	Pavement	Gu et al. 2017 [19]
3	1.0 × 10^−5^	8.0 × 10^−5^	Subgrade	Wang and Zhuang 2021 [20]

**Table 2 materials-16-02141-t002:** Classification of the deformation properties for the investigated samples on the basis of Δ*ε_p_*_,5000_ − Δ*ε_p_*_,3000_.

Degree of Compaction (%)	Confining Pressure (kPa)	Δ*ε_p_*_,5000_ − Δ*ε_p_*_,3000_						
		5%	10%	20%	30%	40%	60%	80%
0.95	40	1.8 × 10^−5^	7.2 × 10^−5^	1.4 × 10^−4^	2.4 × 10^−4^	4.5 × 10^−4^	2.8 × 10^−3^	/
	60	2.6 × 10^−5^	6.9 × 10^−5^	1.4 × 10^−4^	2.6 × 10^−4^	4.7 × 10^−4^	3.2 × 10^−3^	4.6 × 10^−2^
1.00	40	3.2 × 10^−5^	8.5 × 10^−5^	1.6 × 10^−4^	2.6 × 10^−4^	4.3 × 10^−4^	1.9 × 10^−3^	/
	60	4.6 × 10^−5^	1.1 × 10^−4^	2.1 × 10^−4^	3.3 × 10^−4^	5.0 × 10^−4^	3.0 × 10^−3^	/
Classification		Plastic shakedown			Plastic creep		Incremental collapse	

## Data Availability

The data presented in this study are available on request from the corresponding author.

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
