# Peer review of "A New Approach for Classifying the Permanent Deformation Properties of Granular Materials under Cyclic Loading"

_materials, 2023, doi:10.3390/ma16062141_

Round 1

Reviewer 1 Report

The present research investigates the permanent  deformation properties of granular materials with different compaction degrees via medium-sized cyclic triaxial tests under different confining pressures.

The research topic is very interesting and paper is well-written: Introduction states clearly the state of the art and the motivations for present research.

The methods are clearly described and the results consistent. In my opinion, it is acceptable for publication once the authors provide proper statistics for the results. In particular, confidence bounds are required  in figures 11, 13d,14d. Moreover, if a data fitting was performed (Fig.11, 13d,14d) the equation together with the coefficient of determination (R2) must be  provided.

Author Response

A point to point response is attached below.

Reviewer 2 Report

The authors have presented a work on permanent deformation properties of granular material using cyclic triaxial tests. Few suggestion to enhance the work are given below - 

1. Abstract - authors can bring out a specific application for the work 

2. Abstract - results can be presented in terms of numerical values.

3. Introduction - authors can be give a general description of the deformation behaviour of granular material before discussing the shakedown and creep limits.

4. Figure 1 clearly shows particles in sand range too (4.75 mm to 0.75 mm). Why do authors only specify gravel? Granular material would be better choice.

5. Figure 1 shows soil particles of various sizes but in section 2.2, p.4 authors state particles of 31.5 mm were only used. Please explain

6. Why compaction degrees of 0.95 and 1 were selected. The difference in density would not be significant

7. How is the permanent strain rate selected?

8. Conclusion can be a concise summary of the findings highlighting the application and limitations of the study. 

9. New references can be added particularly the investigations conducted in the last five years. 

Author Response

(The authors gave the same response as above.)

Reviewer 3 Report

Dear atuhors

Your paper is really well strucutred and written. There is small room for improvement, just a couple suggestions. Congratulations

Author Response

(The authors gave the same response as above.)
